# Development of Smart Clothing to Prevent Pressure Injuries in Bedridden Persons and/or with Severely Impaired Mobility: 4NoPressure Research Protocol

**DOI:** 10.3390/healthcare11101361

**Published:** 2023-05-09

**Authors:** Anderson da Silva Rêgo, Guilherme Eustáquio Furtado, Rafael A. Bernardes, Paulo Santos-Costa, Rosana A. Dias, Filipe S. Alves, Alar Ainla, Luisa M. Arruda, Inês P. Moreira, João Bessa, Raul Fangueiro, Fernanda Gomes, Mariana Henriques, Maria Sousa-Silva, Alexandra C. Pinto, Maria Bouçanova, Vânia Isabel Fernande Sousa, Carlos José Tavares, Rochelne Barboza, Miguel Carvalho, Luísa Filipe, Liliana B. Sousa, João A. Apóstolo, Pedro Parreira, Anabela Salgueiro-Oliveira

**Affiliations:** 1Health Sciences Research Unit: Nursing (UICISA: E), Nursing School of Coimbra (ESEnfC), 3000-232 Coimbra, Portugal; guilhermefurtado@ipg.pt (G.E.F.); rafaelalvesbernardes@esenfc.pt (R.A.B.); paulocosta@esenfc.pt (P.S.-C.); ltrfilipe@esenfc.pt (L.F.); baptliliana@esenfc.pt (L.B.S.); apostolo@esenfc.pt (J.A.A.); parreira@esenfc.pt (P.P.); anabela@esenfc.pt (A.S.-O.); 2Polytechnic Institute of Coimbra, Applied Research Institute, Rua da Misericórdia, Lagar dos Cortiços–S. Martinho do Bispo, 3045-093 Coimbra, Portugal; 3International Iberian Laboratory of Nanotechnology (INL), 4715-330 Braga, Portugal; rosana.dias@inl.int (R.A.D.); filipe.alves@inl.int (F.S.A.); alar.ainla@inl.int (A.A.); 4Fibrenamics, Institute of Innovation on Fibre-Based Materials and Composites, University of Minho, 4800-058 Guimaraes, Portugal; luisa.arruda@fibrenamics.com (L.M.A.); ines.moreira@fibrenamics.com (I.P.M.); joaobessa@fibrenamics.com (J.B.); rfangueiro@det.uminho.pt (R.F.); 5Centre for Textile Science and Technology (2C2T), University of Minho, 4800-058 Guimaraes, Portugal; chelnebmg@gmail.com (R.B.); migcar@det.uminho.pt (M.C.); 6CEB—Centre of Biological Engineering, LIBRO—Laboratório de Investigação em Biofilmes Rosário Oliveira, University of Minho, 4710-057 Braga, Portugal; fernandaisabel@ceb.uminho.pt (F.G.); mcrh@deb.uminho.pt (M.H.); m.silva@ceb.uminho.pt (M.S.-S.); a77645@alunos.uminho.pt (A.C.P.); 7LABBELS—Associate Laboratory, 4710-057 Braga, Portugal; 8Impetus Portugal-Têxteis Sa (IMPETUS), 4740-696 Barcelos, Portugal; mboucanova@impetus.pt; 9Physics Center of Minho and Porto Universities (CF-UM-PT), Campus of Azurém, University of Minho, 4804-533 Guimarães, Portugal; vaniafernandesousa@gmail.com (V.I.F.S.); ctavares@fisica.uminho.pt (C.J.T.)

**Keywords:** wounds, biomedical technology, anti-infective agents, preventive medicine, pressure ulcers, smart clothing, medical devices, 4NoPressure

## Abstract

Pressure injuries (PIs) are a major public health problem and can be used as quality-of-care indicators. An incipient development in the field of medical devices takes the form of Smart Health Textiles, which can possess innovative properties such as thermoregulation, sensing, and antibacterial control. This protocol aims to describe the process for the development of a new type of smart clothing for individuals with reduced mobility and/or who are bedridden in order to prevent PIs. This paper’s main purpose is to present the eight phases of the project, each consisting of tasks in specific phases: (i) product and process requirements and specifications; (ii and iii) study of the fibrous structure technology, textiles, and design; (iv and v) investigation of the sensor technology with respect to pressure, temperature, humidity, and bioactive properties; (vi and vii) production layout and adaptations in the manufacturing process; (viii) clinical trial. This project will introduce a new structural system and design for smart clothing to prevent PIs. New materials and architectures will be studied that provide better pressure relief, thermo-physiological control of the cutaneous microclimate, and personalisation of care.

## 1. Introduction

Pressure injuries (PIs) include any injuries that can occur due to a person’s organic and physiological context. These factors can combine with unrelieved pressure, shear or friction, and contact with medical devices (MDs) or other equipment used in care, leading to tissue damage or even death [1,2]. The European Pressure Ulcer Advisory Panel (EPUAP) has identified anatomical regions with greater bony prominences to be the most affected areas [3,4,5]. The development and progression of PIs is influenced by the interaction between intrinsic risk factors, such as a patient’s physiological frailty, and extrinsic factors, such as friction and shear caused by the use of MDs [6,7].

PIs are a major concern for patient safety globally, especially among vulnerable populations [8,9,10,11,12]. In the United States of America, more than 2.5 million patients develop PIs every year, with a median prevalence of 15% [13,14,15], while in Europe, the prevalence is 10.8% [3,11]. In Portugal, epidemiological data show a median prevalence of PIs ranging between 11.5% and 17.5% in specific care settings, such as hospitals, long-term care facilities, home care, and residential facilities for older adults [16,17].

The conventional management of PIs involves a combination of pressure-reducing support surfaces and patient repositioning techniques [18]. However, the effectiveness of these methods is debatable. Although some techniques are more effective than others in preventing PIs, no universally effective solutions for preventing or reducing PI occurrence have been clinically validated [19,20]. Therefore, one new approach to managing and preventing PIs involves the integration of conventional and clinical health technology (CHT) solutions [21]. The use of CHT in preventing PIs has been studied to enable early detection of PI development, to reduce the overall risk factors, and to manage tissue burden [22].

There is also scientific evidence recommending the use of potential CHT solutions to assess the risk of a patient developing a PI [23] and to minimise infection risk [24]. These solutions include MDs in developing prototypes that measure tissue load, pressure map monitoring systems, biofeedback of microvascular and biochemical soft tissue properties, and prophylactic dressings, among other treatments [25,26].

The 4NoPressure project proposes the development of a new type of smart clothing based on a novel structural and system concept incorporating a unified matrix of sensors for the monitoring and detection of areas critical for PI occurrence in people with severely reduced mobility and/or who are bedridden [27,28]. Additionally, the project aims to study the use of fibres with distinct and complementary properties that act on the reduction of the coefficient of friction and maintenance of microclimate through the use of phase-change materials (PCMs) incorporated into the fibres. To the best of our knowledge, there is a lack of MDs (or similar) that possess the innovative properties such as microclimate thermoregulation control and antimicrobial properties, among others, that correspond to the clinical needs for PI prevention [27,28].

In recent years, several technologically advanced functional textiles for PI prevention have emerged [29,30]. Smart Textiles for Healthcare (STH) have emerged as a solution for application in various healthcare settings [31]. In the case of PI management, STH solutions are primarily designed to preserve the proper microclimate around the patient’s skin [30]. Despite the possibility of applying this type of fabric for other purposes, studies addressing and validating the STH used to develop clothing to prevent PIs are still in their early stages [29,30]. Therefore, new MDs need to be developed that focus on preventing and reducing the occurrence of PIs and the complications arising from them [32].

The relevance of developing this type of solution becomes even more apparent, when considering the preventions of PIs in people with severely reduced mobility and/or who are bedridden, and the multi-level economic impact that PIs represent with respect to health services in both hospital and home environments. The 4NoPressure project arose to address this market gap based on five key technology points, which translates into the lack of an integrated and functional clothing solution to prevent PIs.

## 2. Objectives

The overall goal of the 4NoPressure project is to develop a new type of smart garment for people with reduced mobility and/or who are bedridden in hospital or home settings within the framework of the concept of SHT-integrated solutions. This garment is intended to be a medical device capable of reducing the occurrence of PIs arising from therapeutic activities, preventing associated infections. In addition, the smart garment is expected to exert a preventive function, monitoring important risk factors in the development of PIs.

Two types of clothing will be developed with distinct purposes and characteristics. Typology A will employ advanced fibrous and bioactive materials but will not incorporate sensor technology. The aim of this device will be to provide a suitable option for daily use in clinical and domestic environments.

In contrast, Typology B will combine advanced fibrous and bioactive materials with sensor technology. This sensory aspect of the clothing will be directed toward the evaluation and monitoring phases of critical patients, such as those in intensive care units and patients with chronic diseases. In this paper, several research and development (R&D) activities [33] for product development will be presented.

## 3. Methods

### 3.1. Project Design

The 4NoPressure project is structured around Research, Development, and Innovation (RDI) activities applied to HSC [34]. Specifically, it is characterised as being contemporary translational research, defined as the effective use of scientific knowledge to develop new health technologies [35]. Furthermore, developing this MD (smart garment) will involve researchers from distinct areas of clinical, technological, and industrial expertise in a UCD approach [36].

### 3.2. Phases of Product Development

To accomplish the development of both types of clothing for the prevention of PIs, several stages will be implemented by academic, technological, and industrial partners (Table 1).

The Technology Readiness Level (TRL) will be used throughout the project’s 8 distinct phases, considering its importance in ensuring that each phase is clearly defined in terms of objectives and success criteria. During each phase, the TRL will be employed to assess the readiness of the developed technology and to determine whether it is progressing towards the defined project objectives [37,38,39,40,41].

In the case of the described project, the TRL will be implemented in all phases to evaluate the readiness of the developed technology. In the first phase, for example, the TRL will be used to assess the readiness of the defined product and the process requirements and specifications. The TRL will be used to evaluate the readiness of the structural fibre system and the textile technology developed in the second phase. In the third phase, the TRL will be used to assess the readiness of clothing design and modelling [41,42,43,44].

Throughout the project, the TRL will also be used to evaluate the readiness of sensor monitoring, bioactivity, and clothing manufacturing processes developed in phases four, five, and six. In the seventh phase, the TRL will be employed to evaluate the readiness of the developed and optimised prototypes. In contrast, in the eighth and final phase, the TRL will be used to assess the readiness of the validated prototypes [45,46].

It is important to note that Human-Centred Design (HCD) should be integrated into all project phases to ensure that the developed technology meets user needs and is technically and economically viable. The integration of HCD and TRL should be clear and constant in all project phases to ensure a successful and satisfactory outcome [37,38,39,40,41,44,45].

#### 3.2.1. Phase 1: Product and Process Specifications

The first phase is critical for identifying clinical indicators that can be used in the development of a new product. A scientific literature review will be performed to identify the epidemiological and clinical outcomes that can be used to guide the initial procedures for developing innovative smart clothing. Additionally, a qualitative research study based on the UCD approach will be conducted, using semi-structured guiding questions, addressing the needs of end users who are people with reduced mobility and/or bedridden, and stakeholders such as formal and informal caregivers and healthcare professionals. An observational study will be conducted in long-term care units, nursing homes, and at-home contexts, involving people with limited mobility and/or who are bedridden to clarify the most important aspects in the development of the textile device.

#### 3.2.2. Phase 2: Development of the Structural Fibre System and Textile Technology

In this phase, three activities will be conducted sequentially. The first activity aims to evaluate different natural, semi-synthetic, and synthetic fibres, individually or in combination, regarding the fulfilment of the thermoregulation requirements. The research team will employ the fibrous architecture and surface design method. The second activity focuses on inserting the evaluated fibres into different knitted structures, such as jersey, piquet, and two other structures developed by an industrial partner. These textile structures aim to contribute to the thermoregulation properties, and the surface designed considering the distribution of the pressure exerted by the body. The third activity involves analysing possible fabric finishing processes, such as fabric softeners, to create a smoother surface on the fibres and to reduce the static and kinetic friction coefficient. Samples of knitted fabrics in different structures will be assessed with respect to their structural, physical, chemical, and mechanical properties.

The study of the surface design of the clothing will provide inputs for the redistribution of localised forces and the dissipation of the concentration of mechanical energy, thus minimising the effects of structural forces. This is particularly important with respect to the prevention of PIs due to decreased blood circulation resulting from pressure on bone prominences. The textile surface’s reliefs are reinforced by two-dimensional and three-dimensional supports in order to absorb the desired impacts.

#### 3.2.3. Phase 3: Clothing Design and Modelling

The first task in this phase involves garment model development and anthropometric characterisation of the target population, including modelling characterisation, safety design, and handling. Anthropometric characterisation of the target population will also be performed using 3D body scanning technology and the measurement of the variables defined for the study, relative to the modelling, that result in configurations achieving better comfort and easy usability.

The results of the first task will inform the second task, where investigations into flat and 2D prototype projections will be carried out. Concepts of design and structural systems will be considered to ensure ergonomic comfort and ease of use. The modelling process will follow criteria for proper movement and avoid excess fabric in order to prevent folds and internal volumes. Additionally, solutions for accommodation will be considered, such as the use of diapers and easily accessible openings for medical devices and hygiene.

The project will use 2D flat modelling on paper, 3D modelling on a mannequin, and 3D Computer-Aided Design (CAD) modelling. The latter allows the performance of several simulations for the digital identification of the pressure and compression points responsible for discomfort caused by wounds leading to PIs in the population under study. The simulation of 3D avatars in different anatomical positions will validate the flat modelling developed for static positions, underscoring the need to change traditional modelling methods.

The perception-related aspects of clothing, characterised by ergonomics, will be taken into consideration with respect aesthetic quality. Subjective aesthetic evaluation is relevant to users, involving aspects that raise self-esteem, confidence, and satisfaction, even in a hospital environment. In this project, essential categories of materials, design, and influencing factors will be identified to support comfort. The third task in this phase will be to build moulds for cutting the textile raw materials and to develop a procedure to build validation models in the laboratory. This task will be carried out in parallel with the activities planned as part of phase 6.

#### 3.2.4. Phase 4: Development of Sensing Technology and Its Connection with Clothing

The activities undertaken in this phase will be carried out as part of five distinct tasks. In the first task, the conductive fibrous matrix for integration into the sensing system will be evaluated. This will involve studying the function of fibre characteristics, exploring coating techniques, and replicating the process on a large scale in the industrial process. These studies aim to identify the need for specific processes in order to obtain the desired functionality for the fibres using low-cost, sustainable materials and expedited processing.

In the second task, centred on the development of conductive fibre structures, functionalisation using chemical solutions (knife-coating, screen-printing, and dip-pad-dry) will be studied, along with the formation of new composites using carbon-based materials as fillers in order to ensure the safety of the sensing platform for end users. The developed sensing textile substrates will be analysed with respect to the stability of their properties as a function of pressure and their conductivity/resistivity, mobility, amount, and type of carrier. A group of techniques will be applied for structural characterisation and to determine chemical composition in order to uncover how the studied parameters influence the electric properties of the sensors. The sensing structures’ capacity for variation and electrical behaviour in response to external mechanical stimulus will be explored on the basis of capacitive and piezoresistive principles.

In the third task, the transduction principles and manufacturing methods will be investigated in light of critical technical knowledge in the different domains of sensor design. As part of this task, the pre-selected sensors’ sensing principles will be further studied, the different types of transductions will be evaluated and selected, and preliminary tests will be conducted to validate the sensors. Programming platforms will be used to implement analytical models already existing in the literature and adapted to smart clothing. To analyse physical and mechanical performance, simulations will be performed using analytical modelling and finite element modelling in the Multiphysics software package.

In the fourth task, technology for integrating the sensors during the product’s manufacturing process will be investigated, in parallel with the first two tasks of this phase. The activities will start by investigating the compatibility of the sensor integration techniques in fibrous materials, in consideration of the textile developed in the second phase. The experimental testing of the sensors will use the printing technique to incorporate the fabric monitoring system using industrial textile manufacturing equipment.

The fifth task will comprise the development of individual sensors, and the testing and validation of setups according to their specifications and responses to stimuli (pressure, temperature, and humidity). Consequently, new fabrication techniques, experimental tests, and readout setups will be developed on the basis of the theoretical underpinnings of the fabrication and testing of individual sensors. The construction of an interface platform for powering and reading the sensors is also envisaged as being part of this task.

#### 3.2.5. Phase 5: Development of Bioactive Technology

This phase consists of four tasks focused on the development and characterisation of photocatalytic nanomaterials and their optimisation for the creation of innovative functional textiles with antimicrobial properties.

The first task will involve synthesising and characterising the photocatalytic nanomaterials using a colloidal solution. Parameters such as the concentration and volume of reactants, reaction time, temperature, doping, type of agitation, and product washing will be checked. Transmission and scanning electron microscopy will be used to analyse the homogeneity, coalescence, and morphology of the material. The surface area and porosity will be determined, followed by crystallographic, structural, and morphological characterisation of the photocatalytic nanomaterials using several methods.

In the second task, the focus will be on optimising the electromagnetic aspects of nanoparticle activation. Different approaches, such as hydrothermal synthesis and heat treatment, will be used to stabilise titanium dioxide doping with metallic and non-metallic elements. These tests will be performed with the aim of overcoming the limitation of titanium dioxide regarding its activation by radiation, considering that photocatalytic nanomaterials absorb light from sunlight or lamps with an electromagnetic spectrum. This task aims to optimise the doping and lighting conditions for clinical trials planned in phase eight.

To ensure the material’s bioactive properties, the third task will be the evaluation of the antimicrobial activity of the textiles using the standard JIS 1902:2008 (E) to test their antibacterial activity and efficacy on textile products. This analysis will incorporate a set of qualitative (e.g., halo measurement) and quantitative (e.g., adsorption method) tests. The cytotoxicity of the antimicrobial agents will be evaluated using the calorimetric tetrazolium microculture method. In vivo tests will be performed to determine point and cumulative skin irritation and sensitivity.

The fourth task will be to assess the resistance of textiles after washing and sterilisation processes at the laboratory level to ensure long-lasting bioactive properties associated with functional textiles.

#### 3.2.6. Phase 6: Clothing Manufacturing Process

First, the conditions required for the production and development of new products will be explored. The layout will be evaluated to determine any necessary changes in the manufacturing process with respect to the yarn reception, knitwear production, seamless technology, finishing cutting, and sewing phases. The final task is to validate the optimisation of the parameters of the productive processes based on the previous tasks, during which the changes will be designed, dimensioned, calculated, and further documented.

#### 3.2.7. Phase 7: Development and Optimisation of Prototypes

In this phase, the semi-functional prototypes will be developed. The typology A prototype will focus on daily/regular use, and will consist of technology related to the advanced fibrous and bioactive materials, without the sensing technology. The typology B prototype will comprise the bioactive materials and the sensing platform.

If necessary, their end users will evaluate these prototypes to determine potential changes before the delineation of the final prototypes.

#### 3.2.8. Phase 8: Validation of Prototypes

A pre-clinical observational study will be performed to validate the final prototypes according to Strengthening the Reporting of Observational Studies in Epidemiology (STROBE) [46] guidelines. Initially, the study will be conducted with a restricted group of end users without pressure injuries. The objective is to improve the product in regulatory terms and to meet the technical specifications and needs established by end users and relevant stakeholders. Additionally, a plan will be established for a clinical investigation with clothing intervention for healthcare (considered a medical device), including validation of safety and efficacy results, determination of sample size, and real-world implementation [47].

In the future, Randomised Clinical Trials (RCTs) [48] will be conducted in which the participants will be randomly assigned to treatment and control groups to evaluate the efficacy and safety of the prototype pyjamas. RCTs are considered the gold standard for evaluating the effectiveness of new types of sleepwear. In this study, participants will be selected based on well-defined inclusion and exclusion criteria and randomly assigned to one of the study groups. The treatment group will receive the new prototype pyjamas being tested, while the control group will receive the existing standard pyjamas. This will allow for a fair comparison between the groups and minimise the possibility of bias [46,47].

The Consolidated Standards of Reporting Trials (CONSORT) [48] guidelines will be used extensively in order to report the results of the RCTs clearly, completely, and transparently. All necessary elements for their implementation, including inclusion and exclusion criteria, data collection and analysis procedures, case report forms, investigator’s brochure, protected consent terms, and all protected documents for implementing the clinical investigation with the pyjama prototypes in a real-world setting will be followed.

Clinical surveillance will be an important aspect of conducting RCTs in the future. This will be performed by a competent subcontracted entity under Article 22 of the Clinical Research Law (Law No. 21/2014) [49]. This will include preparing and submitting the necessary documentation to competent entities, consultation within the scope of good clinical practice, product certification support, and the CE marking process. Additionally, all ethical aspects involved in the research will be carefully studied and defined, such as the standards of good clinical practice and the provisions of the Declaration of Helsinki [48].

### 3.3. Participants’ Involvement

The 4NoPressure project will involve the participation of humans and clinical/medical institutions at different stages. In the first phase, an observational study will be carried out to deepen the analysis of the clinical parameters of the LPs, considering the target group. As part of task two, an observational study will be conducted to collect anthropometric and body composition measurements, as well as clinical and biosocial data among three groups of participants: (i) hospitalised patients; (ii) patients receiving home care; and (iii) long-stay patients in nursing homes and/or long-term care. In this phase, focus groups will also be conducted with formal and informal caregivers and health professionals who assist people with reduced mobility and/or who are bedridden in order to obtain other perspectives on the composition of the pyjamas.

In this stage of observational research, participants will be selected who are 18 years or older, and who are confined to bed and/or in sitting positions (such as wheelchairs and armchairs), who have given informed consent, and who have adequate global scores on the Montreal Cognitive Assessment (MoCA) scale [50], excluding non-collaborative patients. These criteria ensure that the selected participants have similar characteristics and can contribute to the research objectives.

Inclusion criteria for formal and informal caregivers and health professionals who will participate in the study include experience in caring for bedridden or mobility-limited patients, currently caring for a patient at risk of PIs, voluntary participation in the focus group, minimum age of 18 years, and ability to communicate in Portuguese. Those with experience caring for bedridden or mobility-limited patients, not currently caring for patients at risk of suffering PIs, not agreeing to participate voluntarily in the focus group, age under 18, and difficulty communicating in Portuguese will be excluded.

Convenience sampling is a technique that can be used in research when the researcher has access to a specific group of people. Participant selection is based on inclusion and exclusion criteria to ensure sample representativeness and minimise potential biases. The participants in the study will be recruited according to the indication of the nurses responsible for the Community Care Unit. They will be subjected to screening to verify that the nominees fit the inclusion and exclusion criteria of the project. On the other hand, excluding non-collaborative patients is important to ensure that the selected participants are willing to participate in the research and can contribute adequately. This exclusion is important for ensuring that the sample is representative, and the results are reliable.

Thus, choosing a convenience sample based on inclusion and exclusion criteria is an important technique in research aimed at obtaining relevant and representative results. Careful participant selection is crucial for ensuring that the research achieves its objectives and can contribute to our understanding of a particular phenomenon or problem.

Finally, in phase eight, two randomised controlled trials will be conducted for product testing and final validation with health professionals, formal and informal caregivers, and people with reduced mobility and/or who are bedridden. According to the guidelines of the Consolidated Standards of Reporting Trials (CONSORT) [44], each study will follow three phases, which will be detailed in clinical protocols for each target public. These protocols will provide the outcomes, considering usability, fit quality, ergonomic comfort and wellness, modelling characterisation, safety, and handling.

Finally, two randomised controlled trials will be conducted in phase eight for the product testing process and final validation. The expected sample will be convenience-based, with the stipulation of a minimum of 15 participants. The following inclusion criteria will be established: age equal to or greater than 18 years, proficiency in the Portuguese language, healthcare professionals with experience in caring for people with reduced mobility and/or who are bedridden, formal and/or informal caregivers of people with reduced mobility and/or who are bedridden, individuals with reduced mobility and/or who are bedridden who are willing to use the prototype of the pyjamas in a laboratory environment, and who sign their informed consent to participate in the survey. The exclusion criteria will be participants who do not cooperate In the survey and who care for people with no changes in mobility (i.e., who have the ability to walk).

### 3.4. Eligibility

The recruitment of participants and health institutions from diverse contexts will be undertaken through a network of partners involved in the project. Participants who express interest in participating will sign an informed consent form that guarantees the privacy of their identity and any collected data. For hospitals and long-stay facilities, contact with the medical team will be established to verify the eligibility of each subject to participate in the study, according to the previously defined inclusion and exclusion criteria for each phase of the study.

To be admitted to different phases, participants must meet the following inclusion criteria: full communication capacity in Portuguese, age above 18 years, and capability to provide consenting access to their medical report. Those without reduced mobility and/or who are not bedridden will be excluded from the study. Professionals from non-health fields and formal or informal caregivers of individuals without reduced mobility and/or who are not bedridden will also be excluded.

Independent samples from various clinical settings, such as healthcare professionals, nurses, patients, and others, will be recruited by a research team for different studies. Future study protocols will provide details on specific procedures related to eligibility, sample size calculation, blinding, and group allocation, respecting the different phases of the 4NoPressure project [51].

### 3.5. Ethics

The 4NoPressure project involves the development of new smart clothing, which requires a sequence of rigorous operations, including the prototype development and validation process. To ensure compliance with applicable regulations and ethical standards, the project team has taken into consideration European Union directives [52] enforced in Portugal by the National Authority for Medicines and Health Products (INFARMED), as well as guidelines from the International Standard Organisation (ISO) [53]. In addition, the project will follow the procedures established by the Portuguese Ethics Committee for clinical research [54].

The Ethics Committee of the Health Sciences Research Unit: Nursing of the Nursing School of Coimbra has granted central approval for the 4NoPressure project (reference no. 701-07/2020). The project team will conduct clinical studies per the Declaration of Helsinki [52] and the ethical principles that guide Good Clinical Practice.

## 4. Discussion

Preventing and treating PIs is a growing concern for researchers and clinicians, as it represents a major clinical challenge with significant epidemiological impact worldwide [49,50]. PIs affect patients’ quality of life and have psychosocial implications for patients and their families [55]. In hospital settings, PIs are indicators of quality of care. The economic burden of recovering people who develop PIs outweighs the cost of their prevention, highlighting the need to invest in surveillance and monitoring [56]. Recent findings have shown that treatment costs are substantially more expensive than PI prevention strategies, especially when early-stage injuries progress to more advanced stages [57,58].

Innovative strategies that aim to manage the occurrence of PIs can be instrumental in helping those at risk [23]. The current literature points towards a new paradigm for treating and preventing PIs, including both pragmatic and health technology-based solutions, through the combination of different clinical approaches [59,60,61]. Despite the increase in clinical healthcare technology (CHT) innovation applied in PI prevention in recent years [62,63,64], solutions such as interface map pressure, telemedicine, and educational technologies (among others) have demonstrated moderate effectiveness in addressing a subset of risk factors [64,65,66].

The development of new technologies, such as the proposed 4NoPressure project, has the potential to aggregate the current expensive and inefficient approaches used for early-stage PI risk mapping. A recent literature review identified that one of the challenges is the development of technology-based CHT systems that patients can use at home to detect indicators and/or risk factors for PI occurrence and to initiate self-care [23].

With the accelerated progression in the development of miniaturised electronic devices and the global expressive coverage of the Internet, which entails the consumption of digital content, progress in the development of electronic and flexible clothing is promising [67]. Current scientific evidence suggests the directing of research towards the fibres that make up the fabrics of e-wear, on the basis of which the potential of these materials in body applications has been presented [35,68,69].

As preliminary results, the recent findings of the 4NoPressure research group, which performs studies on fibre structure and textile technology, have raised questions related to the search for ways of improving body pressure distribution and thermal regulation of the wearer, which could help prevent the occurrence of PIs.

Advances in the textile industry, coupled with competitiveness and consumer interest, have been shifting towards new product concepts, leading to the engineering of new innovative and smart technologies/textiles with a wide range of applications, where the functional performance represents a key factor [70,71,72].

To develop new medical devices, textiles that balance patients’ thermo-physiological and comfort parameters need to be integrated into a new type of fabric, taking into consideration the particularities of the skin microclimate [67]. Smart textiles have emerged as a support tool for health professionals and the caregivers of people with reduced mobility and/or who are bedridden in order to prevent the occurrence of PIs [73], emphasising the need to continue employing the care practices currently in use, while pyjamas can serve as a tool to help in the work routine.

The 4NoPressure project aims to incorporate materials possessing properties such as thermal conductivity, electrical conductivity, sensing ability, and antimicrobial properties into functional textiles. Electrical conductivity is a key property in order for a fabric to be considered smart, followed by flexibility, and light weight [74]. Fabric properties such as air and water vapour permeability, absorption, wetting and drying must be incorporated into the smart textile research and development process [22,75,76].

The textile industry is known for being heavily regulated regarding environmental impact, yet it remains one of the most polluting industries [77,78]. We aim to merge environmental concerns with social responsibility, taking into consideration the significant socio-environmental impact of the industry from material selection to product disposal [79,80,81].

In the apparel development process, specifications for apparel models will target people with MRI and/or who are bedridden, with a focus on ensuring that their clothing has a longer lifetime. In phase three, specific tasks such as anthropometric measurement, ergonomic comfort, and psychological–aesthetic comfort [29,30,82,83] are essential for ensuring that the piece is properly dimensioned, providing the range and mobility necessary to perform daily activities [30,82,83].

Aligned with textile issues, the inherent flaws in the design and modelling of clothing based on anthropometric data show the importance of having this indicator as one of the pillars in the prevention of PI. The design created on the basis of the information learned in the first phase allows the creation of products that better fit the body and provide greater comfort, reducing the possibility of PI occurrence.

Textile sensors capable of responding to chemical, biological, and mechanical stimuli are critical for smart clothing sensor technology [82,83,84,85]. While studies have advanced in conductive fibres and textile-based fabrics, there is still a lack of conductive fabric-based sensors for determining clinical parameters for bedridden and mobility-impaired individuals [86,87]. The challenge faced by the research team in this project is to balance conductive fibres and electrical properties, which can be reduced when considering the knitting and stretching of the electrodes, potentially interfering with the sensor’s signal capture [88]. The sensor technology being developed by 4NoPressure will allow monitoring and early detection of risk factors for developing the understanding the causes of PIs in consideration of the skin’s microclimate.

Textile sensors need to be sensitive to changes in the skin microclimate [86], while taking into consideration the clinical parameters of individuals susceptible to PIs [30]. One study identified the potential of actuator materials due to their linear and rotational properties. However, energy loss during phase transformation can cause hysteresis behaviour and nonlinear actuation in Shape Memory Alloy (SMA) actuators, limiting their commercial application due to parameter uncertainties and relative costs [84]. An American study has identified particulate and fibrous structures that enhance mechanical, thermal, electrical, and optical signals [85].

The research team must determine which structures best meet the proposed objectives while prioritising end-consumer safety regarding comfort and long-term contact without causing damage to their skin [88]. Antimicrobial functionalisation of textiles has various clinical applications [85]. Textiles are susceptible to microbial action, leading to bad smells, degradation of mechanical properties, discolouration, decomposition, dermal infections, allergic reactions, and conditions resulting from bacterial growth [69,89,90,91]. Functionalised textiles with antimicrobial properties must meet specific standards and regulations to maintain product quality and properties [92].

Bioactive technology with antimicrobial and biodegradable properties, developed by 4NoPressure, can help keep skin hydrated and improve patients’ quality of life. The preliminary results being analysed come from the synthesis of microencapsulation and nanoparticle systems, which need more robust laboratory testing before it can be concluded precisely which synthesis methods and materials can be used in the wearable devices. It is worth mentioning that these issues foresee the activation of nanoparticles through the electromagnetic spectrum and functionalisation concerning antimicrobial and skin moisturising activities and the adhesion process to the developed textile.

Durability, textile carrier selection, preparation, and the application of microcapsules are crucial factors to consider. Textiles with encapsulated antimicrobial agents aim to prevent bacterial growth, control long-term hygienic conditions, reduce unpleasant odours, and maintain a healthy skin environment, making them a critical focus of investigation [93]. Achieving effective antimicrobial activity without cytotoxic effects and ensuring durability after several wash cycles is the key challenge and objective of this project. Natural antimicrobials are being considered due to their health benefits and wide range of properties, which include antibacterial, antifungal, antiviral, insecticidal, antioxidant, anti-stress, soothing, refreshing, and invigorating properties [94,95].

Clinical validation studies of the developed smart clothing can be improved by performing pre-clinical investigations using prototypes A and B. Laboratory assessments may also benefit from specialised assessment by the 4NoPressure Project team for garment design, which could clarify the importance of end user usability [96,97,98,99,100]. The RCT evaluations in the clinical trials phase will validate the intelligent clothing and provide important information about the MD’s behaviour in different clinical and physiological contexts for individuals with reduced mobility and/or who are bedridden. Safety, clinical performance, and effectiveness will be investigated in clinical trials, meeting national and international standards, while always keeping in mind the initially defined needs of end users [98,101,102,103,104,105,106].

To ensure the quality and safety of a medical device, it must go through several development and validation phases. Implementing TRL throughout the project is essential for assessing the readiness of the technology developed at each stage [107,108,109]. In addition, for a medical device to be approved by regulatory agencies, endpoints must be established for each component that ensure their effectiveness, clinical performance, and safety. This means that a clinical investigation plan must be developed, taking into consideration the specificities of the medical device and the needs of patients. In this context, clinical validation through an RCT is essential to provide confidence in the product. The final prototype must incorporate all technologies developed, including textiles, sensor monitoring, bioactive solutions, and manufacturing processes.

Thus, assessing the readiness of the technology developed through TRL and clinical validation through an RCT are fundamental steps in ensuring the effectiveness, clinical performance, and safety of a medical device. Carrying out a clinical investigation plan that foresees specific endpoints for each of these components is a regulatory requirement that must be met for the medical device to be approved and made available to patients [110,111,112].

## 5. Limitations

Before discussing the proposed protocol, it is important to mention possible constraints that may impact its implementation. It should be noted that limited access to certain clinical settings may cause delays and constrain the performance of observational studies aimed at exploring the clinical parameters of the medical device in specific patient groups, as well as the validation of prototypes and final clinical studies. Additionally, the current global shortage of materials and components, especially electronics, which are essential for the development and testing of prototypes, may pose a challenge. Therefore, adjustments will be made to the schedule to ensure compliance with the necessary stages of developing the medical device.

## 6. Conclusions

This study aimed to present the research protocol of a project with the main objective of developing smart clothes to prevent PIs. The results of each phase will provide an opportunity to improve the technical specifications of the medical device based on clinical and applied research and experimental and technical development.

The textile production techniques used to create smart clothing will contribute to new research on intelligent fabrics targeting bedridden people or people with severe mobility limitations for controlling risk factors in the prevention of PIs. By aligning with the manufacturing process, the design of the smart clothing, based on the perceptions of health professionals, end users, and caregivers, will enable better usability, ease of use, and user satisfaction with the device. Technologies with bioactive and antimicrobial properties will foster new perspectives of care beyond traditional techniques like alternating decubitus, skin hydration, and the usual dressings used in PI care.

Despite intrinsic clinical management and resistance to the cleaning/washing process, the products used in the nanoparticles may provide comfort due to the potential product properties being investigated and their possible use. Wearable sensors for skin microclimate monitoring will enable a reduction in shear and friction pressure and thermal and humidity control from a sensor array, with signals encoded through software and available to the caregivers and/or health professionals accountable for the provision of caregiving practices. The clinical trials will ensure that the medical device can be used in the home, as well as in hospital and long-term care or nursing home settings.

The results of this project could facilitate significant advances in smart clothing techniques with a view to achieving integral care, and the planning of innovative educational practices for health promotion and education in the management of the self-care necessary for preventing PIs.

## Figures and Tables

**Table 1 healthcare-11-01361-t001:** Outline of the 4NoPressure Project stages for product development.

Phases	Short Description of Tasks	Scope
Studying and defining requirements and specifications at the product and process levels.	Deepening the analysis of clinical parameters of pressure injuries, considering target groups; defining requirements and specifications of materials, ergnomic parameters, and clothing design; studying specifications of the sensor platform model; defining requirements and specifications of the manufacturing process; studying and regulatory framework of clothing for hospital and home use.	Clinical and applied research
2. Product and Process Specifications	Fibre and textile structure research with three main goals: thermoregulation, better distribution of body pressure, and reducing the coefficient of friction.	Applied research
3. Clothing design and modelling	Design and modelling research based on anthropometric data from the target population. Prototype flat and three-dimensional modelling research.	Applied research
4. Development of sensing technology and its connection with clothing	Study of the wearable textile sensing technology, its electrical properties, and the possibility of integration into reactive sensing systems. Development of individual sensors and test setups.	Applied research
5. Development of bioactivity technology	Development of technologies with bioactive and biodegradable properties. Evaluation of the antimicrobial activity, cytotoxicity, and resistance of bioactivity to the washing.	Applied research
6. Clothing manufacturing process	Study of the manufacturing conditions, including layout, possible changes, and optimisation of production norms and standards.	Applied research
7. Development and optimisation of prototypes	Production of the two prototype versions. Research the semi-functional prototypes to analyse and optimise the product.	Experimental development and research design
8. Validation of prototypes	The final functional prototypes will be tested in pre-clinical studies to analyse whether they meet the previously defined technical and functional requirements. According to national and international standards, clinical research will be implemented.	Experimental development and research design

## Data Availability

The data presented in this study are available upon request from the corresponding author. Data will not be made publicly available due to the patenting process of the device under development.

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
