# Peer review of "Development of Smart Clothing to Prevent Pressure Injuries in Bedridden Persons and/or with Severely Impaired Mobility: 4NoPressure Research Protocol"

_healthcare, 2023, doi:10.3390/healthcare11101361_

Round 1
Reviewer 1 Report
The article describes the process of developing a new type of smart clothing to prevent pressure injuries.The article presents the study of new materials and architecture to provide better pressure relief, thermo-physiological control of the cutaneous microclimate, and personalization of care. The project proposal is presented in the article, but the discussion is lacking some concrete ideas that would increase the article's value.
The main paper's limitation is the discussion. Being a project proposal, it is important to see at least in the discussion the results of a preliminary analysis
Author Response
Corrections suggested by reviewers for health care-2299222 - Development of smart clothing to prevent pressure injuries in bedridden people or people with severely impaired mobility: 4NoPressure Research Protocol.
Greetings to the reviewer. We appreciate the important suggestions and accept all of them as pertinent to the clarity of the study under evaluation.
Changes and arguments are described in topics and by reviewer.
Reviewer 1:
- The article describes the process of developing a new type of smart clothing to prevent pressure injuries. The article presents the study of new materials and architecture to provide better pressure relief, thermophysiological control of the cutaneous microclimate and personalization of care. The project proposal is presented in the article, but the discussion lacks some concrete ideas that would increase the value of the article.
- Answer: We appreciate the contributions and put some expected results, as a form of possibility. As it is a protocol of a study that is being carried out and there is a patent application. In this way, we think we are cautious with the results, especially with those that we have not yet obtained. New arguments have been inserted that reinforce the phases of the project that is under development.
Reviewer 2 Report
PIs are a major concern for patient safety worldwide, especially among vulnerable populations [8]. - do other sources also confirm this? Please refer to other sources
PIs are a major concern for patient safety worldwide, especially among vulnerable populations [8]. - do other sources also confirm this? Please refer to other sources
Please indicate exactly which groups of patients are most at risk of Pressure injuries.
Discussion:
One study identified actuator materials' potential to detect mechanical stimuli but ex- posed their limitations in industrial production, which could cause the material to lose
properties due to the manufacturing process's complexity [80]. Please tell me what the exact test was
In the discussion, please add a comparison of your research results with other researchers. I have the impression that the discussion focuses mainly on own results.
Author Response
Corrections suggested by reviewers for health care-2299222 - Development of smart clothing to prevent pressure injuries in bedridden people or people with severely impaired mobility: 4NoPressure Research Protocol.
Greetings to the reviewer. We appreciate the important suggestions and accept all of them as pertinent to the clarity of the study under evaluation.
Changes and arguments are described in topics and by the reviewer.
Reviewer 2:
- PIs are a major concern for patient safety worldwide, especially among vulnerable populations [8]. - Do other sources also confirm this? See other sources.
- Answer: We appreciate the valuable contributions and other references were cited that support the same argument and that reinforce patient safety, considering the care practices and clinical management of patients, the correct technique and preventive safety tool.
- Indicate exactly which groups of patients are most at risk for pressure injuries.
- Answer: in the fifth paragraph of the introduction, where the objective issues of the 4NoPressure project are described, the target population was inserted. In the text, according to the need, these suggestions were also accepted and described as recommended.
Discussion:
- One study identified the potential of actuator materials to detect mechanical stimuli but exposed their limitations in industrial production, which could cause the material to lose properties due to the complexity of the manufacturing process [80]. Please tell me what the exact test was.
- Answer: The type of material and the phase of action that favors the loss of its properties in the manufacture of the textile were inserted, which can raise the higher costs in the manufacturing process. The answer to his question is between 619-626.
- In the discussion, please add a comparison of your search results with other researchers. I get the impression that the discussion focuses mainly on the results themselves.
- Answer: We appreciate the suggestions approved and accept the questions proposed by the reviewer. In the first paragraph of the discussion, we discuss the issues related to prevention, which is what underlies the construction and development of this medical device.
- In the second paragraph, we point out issues related to the use of new technologies, especially pressure mapping. The third paragraph discusses the literature about the advances in the development of CHT systems, especially for those that can use them at home.
- The fourth paragraph discusses the issues related to the use of wearables and the potential wearable devices.
- Seventh paragraph we discuss preliminary results on the production of the tissue, which is under development. Subsequently, we confront the issue of the use of tools to detect disorders related to the microclimate of the skin.
- In the following paragraphs, we discuss about the comfort, related to the permeability of the fabric and the environmental issues regarding the material used in the manufacture of the prototypes. We discuss the importance of end-user-centered design and the importance of inserting this population into product development. This is accomplished by sensing technology and nanoparticles releasing moisturizing products directly into the skin. Finally, we discuss the issues regarding duration testing and clinical validation, which are important for making the product safe and potentially commercial.
Reviewer 3 Report
First of all, I would like to thank you for the opportunity to review the paper entitled:
”Development of smart clothing to prevent pressure injuries in 2 bedridden or severely impaired mobility persons: 4NoPressure 3 Research Protocol”. In this research, authors are describe a protocol for developing a smart clothing to prevent pressure injuries in 2 bedridden or severely impaired mobility persons. In general, the paper has an interesting and relevant topic for researchers and caregivers, medical staff, and the written level of English is very good and easy to read. However, the end user involvement need to be rewritten. With your permission I recommend the following:
Participants involvement
You describe very well how you will develop your product involving end-users. Did you think to involve also the caregivers and tertiary users? Developing such an interesting tool for receiving feedback's from the companies or national agency, the actors that can buy your product or disseminate is an important step in your methodology. Going forward, I didn't find any argument for using MoCA questionnaire for your inclusion/exclusion criteria. Using only one questionnaire (determine the level of MCI and AD) and informed consent is a limitation of your protocol. Maybe you should check other surveys for constructing a protocol that can have 3 phases (starting the pilot test, during the pilot test, in the end of piloting test), including an well-being questionnaire, etc In the end the end users should improve their physical and psychological conditions, the caregivers start to have more time for themselves and the tertiary group are buying your product - one of the aim of your project. Good luck in the implementation.
Author Response
Corrections suggested by reviewers for health care-2299222 - Development of smart clothing to prevent pressure injuries in bedridden people or people with severely impaired mobility: 4NoPressure Research Protocol.
Greetings to the reviewer. We appreciate the important suggestions and accept all of them as pertinent to the clarity of the study under evaluation.
Changes and arguments are described in topics and by the reviewer.
Reviewer 3:
Participant involvement
- You describe very well how you will develop your product involving end users. Have you thought about involving caregivers and tertiary users as well?
- Answer: there will be the participation of formal and informal caregivers and health professionals who work in the care of people with reduced mobility and/or bedridden. In the participant involvement phase, only the target patients were mentioned, but there will be rounds of focus groups with caregivers and health professionals. Because it is a general protocol, the participants of the Randomized Clinical Trial were not described, but all the individuals questioned by the reviewer will be present in this evaluation phase. Anyway, this description was inserted considering the suggestions made.
- Developing such an interesting tool to receive feedback from companies or national agencies, from the actors who can buy your product or promote it is an important step in your methodology. In the future, I haven't found any arguments for using the MoCA questionnaire for your inclusion/exclusion criteria. Using only one questionnaire (determining the level of MCI and AD) and informed consent is a limitation of your protocol.
- Answer: We appreciate the comments and believe that the suggestions will make the study clearer. It was inserted that for observational studies, people with reduced mobility and/or bedridden, formal and informal caregivers and health professionals will participate. Both in the early stages and in the phases of clinical trials.
- Perhaps you should consult other research to build a protocol that can have 3 phases (beginning of the pilot test, during the pilot test, at the end of the pilot test), including a wellness questionnaire, etc.
- Answer: we entered some results that will be used, but the questionnaire was not described due to the need to follow the making of the pajamas and build the questionnaire precisely, considering the three-phase process. The suggestion of the pilot test was accepted. We reiterate that the pilot test protocol is being developed and will feature the questions pointed out by the reviewer. Thank you for this pertinent information.
- In the end, end users must improve their physical and psychological conditions, caregivers have more time for themselves, and the tertiary group is buying their product - one of the goals of their project. Good luck with the implementation.
- Answer: because it is an innovation/creation study, the implementation of the project will be done in another protocol, planned when the product is validated and approved by the competent health authorities.
Reviewer 4 Report
1. The wording should be changed on line 327.
"Write a text about the selection of the sample for convenience based on inclusion and exclusion criteria"
2. The possibilities of promoting new perspectives of skin care are explained and that health education is carried out to be able to use these techniques, I would recommend you to be more careful so that it is not confused with the substitution of movements to avoid ulcers by pressure, since this physical force (pressure) cannot be solved with the clothes that are proposed.
Author Response
Corrections suggested by reviewers for health care-2299222 - Development of smart clothing to prevent pressure injuries in bedridden people or people with severely impaired mobility: 4NoPressure Research Protocol.
Greetings to the reviewer. We appreciate the important suggestions and accept all of them as pertinent to the clarity of the study under evaluation.
Changes and arguments are described in topics and by the reviewer.
Reviewer 4:
- The wording should be changed in line 327. "Write a text about the selection of the sample for convenience based on the inclusion and exclusion criteria."
- Answer: We appreciate the suggestion regarding recruitment, and selection of participants, which was not described in the participant's engagement. We have taken up what has been suggested and inserted these questions into the text.
- The possibilities of promoting new perspectives of skin care and health education to be able to use these techniques are explained, I recommend that more care be taken so that it is not confused with the replacement of movements to avoid pressure ulcers, since this physical force (pressure) can not be solved with the clothes that are proposed.
- Answer: We appreciate the sensitivity that the device is an auxiliary tool, which does not replace the care practices of health professionals. In the second paragraph of the conclusion, this information has been reinforced, and we have changed any sentence in the text that alludes to these concerns.
Round 2
Reviewer 1 Report
The new arguments have increased the paper's contribution despite the need for more details.